# Climate Variables Related to the Incidence of Human Leishmaniosis in Montenegro in Southeastern Europe during Seven Decades (1945–2014)

**DOI:** 10.3390/ijerph20031656

**Published:** 2023-01-17

**Authors:** Sanja Medenica, Nataša Miladinović-Tasić, Nikola M. Stojanović, Novak Lakićević, Božidarka Rakočević

**Affiliations:** 1Public Health Institute Podgorica, 81110 Podgorica, Montenegro; 2Department of Microbiology and Immunology, Faculty of Medicine, University of Niš, 18000 Niš, Serbia; 3Center of Microbiology, Public Health Institute Niš, 18000 Niš, Serbia; 4Department of Physiology, Faculty of Medicine, University of Niš, 18000 Niš, Serbia; 5Clinical Center of Montenegro, 81110 Podgorica, Montenegro

**Keywords:** leishmaniosis, Montenegro, climate conditions, temperature

## Abstract

Leishmaniosis (or leishmaniasis) is a neglected parasitosis most commonly transmitted by the sandfly bite. Changes in temperature, precipitation, and humidity can greatly affect the vectors and reservoir hosts. This study aimed to determine the association between temperature, air humidity, and weather conditions with the incidence of leishmaniasis in Montenegro during a seven-decade period (1945–2014) and to statistically compare and correlate the obtained data. In the studied period, there were 165 registered cases of leishmaniosis, 96.4%, in the coastal and central region of Montenegro, with an average incidence rate of 0.45/100.000. The visceral form of leishmaniosis predominated (99% of the cases), with only one case of cutaneous disease. Climate factors (average temperature, air humidity, and precipitation) had an impact on the occurrence of leishmaniosis in Montenegro. Air temperature elevated by 1 °C in all regions of Montenegro was significantly correlated with an increased incidence of leishmaniosis, by 0.150 (0.013 to 0.287; *p* < 0.05). In order to improve prevention and control of this disease, it is also necessary to investigate other factors with a possible impact on the number of cases of this neglected parasitosis.

## 1. Introduction

Leishmaniosis (or leishmaniasis) is a parasitic disease which belongs to the category of neglected tropical diseases. It is estimated that ca. 700,000 to a million new cases occurs globally per year [1,2]. This disease is caused by the protozoa from the Leishmania genus, an obligatory intracellular parasite causing three clinical disease entities [3,4]. If left untreated, visceral leishmaniosis (VL), also known as *kala*-*azar*, is fatal in over 95% of cases. It is estimated that 50,000 to 90,000 new cases of VL occur worldwide every year, and only 25–40% of cases are reported to the WHO. Cutaneous leishmaniosis (CL) is the most common disease form, which is manifested as skin lesions which are associated with a serious degree of disability or stigma. In 2020, over 90% of new cases of VL and 85% of CL cases reported to the WHO came from 10 countries. The third entity involves mucocutaneous leishmaniosis which leads to the partial or complete destruction of the nose tissue, oral cavity, and throat mucosa [2]. Numerous animal species are natural reservoirs of the infection [5]. Small mosquitos from the Phlebotomus or Lutzomyia genus serve as parasite vectors [4,6]. Human leishmaniosis is primarily an anthropozoonosis and in addition to the vector-mediated transmission, it can be transmitted by blood transfusion, sharing needles, organ transplantation, or sexually, transplacentally (congenital leishmaniosis), and through hemodialysis [6,7].

The prevalence of leishmaniosis is determined by the presence of phlebotomus on a territory, and by longer periods of warm and dry weather, i.e., in moderate, subtropical, and tropical climates. Leishmaniosis most commonly spreads through a female sandfly bite, and that is the reason why climate changes have a direct impact on the disease spread, for instance, the effect of temperature on the sandfly development or, indirectly, by the effect on the environment upon which the seasonal prevalence of this vector species depends. Regarding the biosphere, geographical variables of the transmission of leishmaniosis are associated with tropical zones with warm and wet climates and regular rainfalls [1,8,9,10]. Knowledge of the geographical and environmental variables is vital for understanding the causes of epidemics and disease transmission, both globally and regionally [1,9,11]. The occurrence of other vector-borne diseases, such as Schistosomiasis and Denga, is known to follow the pattern of climate change as well [12,13,14].

Changes in the temperature, precipitation, and humidity can have major effects on vectors and host reservoirs, changing their distribution and influencing their survival and population size. Small temperature fluctuations can have an impact on the developmental cycle of Leishmania spp. promastigotes in sandflies, enabling parasite transmission into previously non-endemic areas for leishmaniosis. Climate changes are associated with human migrations into the regions where leishmaniosis is endemic, and due to droughts and/or floods, they can cause famine which may disturb the human immune status [2]. In accordance with the proposed climate changes in the Montenegro area is the recent publication predicting an increase in temperature over the decades to come, which would impact the occurrence of vector-mediated transmissions and cause an increase in disease prevalence [15].

This study aimed to determine the association between temperature, air humidity, and weather conditions with the incidence of leishmaniosis in Montenegro during a seven-decade period (1945–2014).

## 2. Materials and Methods

This study involved the people affected by leishmaniosis (or leishmaniasis) in the territory of Montenegro from 1945 to 2014. Montenegro is located in southeastern Europe, with a population of 620,045 [16,17]. Owing to its geographical position, there are several climate zones in Montenegro (the Mediterranean, the modified Mediterranean, and the temperate continental climate).

Information about the disease patients was obtained from the database of the Center for Disease Control and Prevention, Public Health Institute of Montenegro, epidemiological surveys undertaken by the appropriate departments of the primary health care centers, and from the patients’ medical records [18,19]. Parasitological and serological diagnosis of leishmaniosis was performed at the laboratory of the Center for Medical Microbiology, Public Health Institute of Montenegro.

For the period of observation of leishmaniosis, meteorological data on average atmospheric temperatures, relative air humidity, and precipitation were taken from the Center for Hydrometeorology Seismology of Montenegro (Podgorica, Montenegro). The data described each month in a year for the northern (highlands), central, and southern (coastal) regions of Montenegro [19].

Statistical data processing was performed using the SPSS software package version 18.0 (SPSS Inc., Chicago, Illinois) and Joinpoint Regression Program version 4.2.0.2 (Statistical Methodology and Application Branch, Surveillance Research Program, US National Cancer Institute). Associations between the number of patients, calendar months, and meteorological factors were examined using univariate and multivariate linear regression analysis. The values of the regression coefficient (B) and their 95% confidence intervals (95% CI) were calculated. Statistical significance of B values was assessed using the t-test.

## 3. Results

In the studied period, 165 individuals with leishmaniosis were registered in Montenegro (Appendix A). The coastal and central regions accounted for 96.4% of the affected by this parasitic disease. The average incidence rate of leishmaniosis in the observed period was 0.45/100.000 (2–3 affected individuals per year). Geographical distribution of the patients indicated the presence of leishmaniosis in 14 municipalities (out of 21 in total), and most of the diseased were reported in Podgorica—central region (52 patients, with an average incidence rate of 0.66/100.000); Bar—coastal region (46 patients, with the average incidence rate of 2.15/100.000); and Ulcinj—coastal region (16 patients, with the average incidence rate of 1.25/100.000). In 99% of the affected, the visceral form of leishmaniosis predominated, with only one case of cutaneous disease form (Figure 1).

Univariate regression analysis demonstrated that each one-degree temperature increase in the coastal region was associated with a significant rise in the number of people affected by leishmaniosis in March, April, and July, and with the yearly average by 1.068 (0.117 to 2.019; *p* < 0.05), and the same occurred in the central part and highlands of Montenegro (Table 1).

Each air humidity elevation by 1% was significantly correlated with a significant rise in the number of people affected by leishmaniosis in the coastal region, in March by 0.225 (0.021 to 0.429; *p* < 0.05), and in October by 0.271 (0.070 to 0.473; *p* < 0.01), while in the central part, the number of those affected was elevated only in December. In April, May, June, July, August, and September, the number of those affected by leishmaniosis dropped. In January, May, and November in the highlands of Montenegro, each air humidity elevation by 1% was significantly correlated with a rise in the number of those affected by 0.184 (Table 2).

Each precipitation increase of 1 mm was correlated with a significant rise in the number of people affected by leishmaniosis in coastal Montenegro in January, February, June, and December, and at the yearly precipitation level by 0.003 (0.000 to 0.005; *p* < 0.05), with a significant reduction of the number of cases in August—by 0.035 (−0.005 to −0.064; *p* < 0.05). In central Montenegro, the number of cases of the disease significantly increased in February, June, and December, and significantly decreased in April and August.

The number of cases of leishmaniosis increased in the highlands of Montenegro in February and May, and decreased in the same region in June, August, and November (Table 3).

Using multivariate regression, air temperature was confirmed as a meteorological factor significantly correlated with the number of people affected by leishmaniosis in the whole territory of Montenegro (all three studied regions). A temperature elevation of one degree was associated with a significant rise in the number of disease cases—0.150 (0.013 to 0.287; *p* < 0.05) (Table 4). The regression model containing the mentioned three factors and the regression constant (B = −4.039) was able to explain 16% of the variance in the number of disease cases (determination coefficient R^2^ = 0.160).

A significant correlation between the number of cases of leishmaniosis and meteorological factors (temperature, humidity, and precipitation) was not confirmed in coastal Montenegro (data not shown), while air temperature and humidity in central parts of Montenegro were significantly associated with a higher number of cases of the disease. Air temperature elevation of one degree was associated with a significant rise in the number of disease cases—by 1.178 (0.421 to 1.935; *p* < 0.01), while air humidity elevation by 1% was associated with a drop in disease cases by −0.314 (−0.029 to −0.599; *p* < 0.05) (Table 5). The regression model with the above three factors and regression constant (B = 3.103) was able to explain 22.5% of the variance in the number of disease cases (determination coefficient R2 = 0.225).

In the highlands of Montenegro, the temperature was confirmed as a meteorological factor significantly associated with the number of cases of leishmaniosis (Table 6). Air temperature elevation by one degree was associated with a significant rise in the number of disease cases—by 0.174 (0.028 to 0.319; *p* < 0.05) (Table 6). The regression model with the above three factors and regression constant (B = −5.552) was able to explain 9.4% of the variance in the number of disease cases (determination coefficient R2 = 0.094).

## 4. Discussion

In addition to the latitude and altitude, the climate in Montenegro is determined also by the presence of large bodies of water (the Adriatic sea, Lake Skadar), deep-sea inlets (the Bay of Kotor), moderately high mountainous hinterland close to the coast (Orjen, Lovćen, Rumija), Ulcinj field far in the southeast, and the mountain massifs of Durmitor, Bjelasica, and Prokletije. The coastal (southern) and central regions of Montenegro (the Zeta—Bjelopavlići plain) are areas with a Mediterranean climate with long, hot, and dry summers and relatively mild, rainy winters [20]. Settlements in the valleys, such as Podgorica and Danilovgrad, have in January lower temperatures than coastal settlements at approximately the same latitude, while in the summer they have slightly higher temperatures. The farthest north (highlands) of Montenegro has the continental climate type, characterized by low annual precipitation (fairly evenly distributed by months), in addition to significant daily and yearly temperature amplitudes. In the highlands, summers are relatively cold and wet and winters are long and sharp, with frequent frosts and low temperatures (which steeply drop with increasing altitude) [16]. The central part of Montenegro has the greatest number of summer days, with an average yearly number of about 129 days). The coastal region of Montenegro has 100 to 110 summer days and the highlands between 8 and 40 summer days. A summer day is a day when the air temperature reaches 25 °C or more [21].

Montenegro has very rich flora and fauna, as well as a very diverse ecosystem. It is considered one of the most diverse floristic regions in the Balkan Peninsula. The total proportion of protected areas in the national territory is 9.21%, which mostly refers to five national parks [22]. From 2000 onwards, a program of biodiversity monitoring has been implemented, primarily focused on most representative species and habitats of both national and international significance. Forest vegetation is exposed to the greatest pressure due to incessant exploitation. Coastal ecosystems are also exposed to significant risks due to the transformation of natural habitats into human settlements and even urban areas. Due to various pollution sources, water ecosystems are also exposed to substantial pressures, which reduce their productivity [22]. The functioning of many of the ecosystems is also adversely influenced by climate change, the action of which is nowadays one of the most pressing problems [23].

Leishmaniosis is a climate-sensitive disease, since the reproduction and behavior of its vector, the sandfly, is under strong impact of meteorological parameters: precipitation, air temperature, and humidity [24,25,26,27]. Our study demonstrated that per every degree of air temperature rise in the coastal and central region yearly, there was a statistically significant positive correlation with the increase in the number of people affected by leishmaniosis. Numerous studies have shown that temperature impacts both the development of leishmania as an infective parasite in sandflies [28,29] and the life cycle of its vectors—sandflies—with an impact on oviposition, defecation, hatching, and proportion of the presence of adult species [30,31,32]; in other words, increasing temperatures can lead to increased mortality of adult disease vectors [31]. The temperature (maximal, minimal, and mean) is the key factor for both sandflies and disease distribution [33,34,35,36,37,38].

In the coastal region, every increase in air humidity of 1% is significantly associated with an increased number of people affected by leishmaniosis, while in the central region the number is higher in December and lower from April to September. In the highland region of Montenegro, every increase in air humidity of 1% is significantly associated with an increased number of people with leishmaniosis. In the coastal region of Montenegro, every increase in precipitation (mm) of 1 mm was significantly associated with the increased number of people with leishmaniosis in January, February, June, and December, and the level of yearly precipitation by 0.003 as well. This study also showed that an increase in precipitation of 1 mm was associated with a significant reduction in the number of affected people in August (by −0.035). In the central part of Montenegro, every increase in precipitation of 1 mm was significantly associated with the increased number of people with leishmaniosis in February, June, and December, while in April and August, the number was significantly lower. In the highland region of Montenegro, increased numbers of the affected were seen in February and May and reduced in June, August, and November.

Multivariate regression analysis did not confirm any significant correlation of the number of people affected by leishmaniosis with meteorological factors (air temperature, humidity, and precipitation) in coastal Montenegro during seven decades of human leishmaniosis prevalence monitoring. In the central region of Montenegro, with average yearly values of observed meteorological factors, it was established that air temperature and humidity were the climatic factors significantly associated with the number of people affected by leishmaniosis. An air temperature rise of 1 °C was associated with a significant rise in the number of people affected by the disease (by 1.178), while air humidity rise of 1% was associated with a drop in the number of the affected by −0.314. In the highland region of Montenegro, in the observed period, air temperature rise of 1 °C was associated with a significant rise in the number of the affected (by 0.174).

In the whole territory of Montenegro, with average yearly values of meteorological factors (air temperature, humidity, and precipitation), the temperature was significantly correlated with the number of people affected by leishmaniosis. An air temperature rise of 1 °C was associated with a significant rise in the number of the affected (by 0.150).

Many studies have shown that climatic variables have an impact on the incidence of leishmaniosis. The impact of specific climatic factors can be both positive and negative, depending on the geographical characteristics and season [39]. In a study in Bangladesh, the authors analyzed the impact of air temperature, humidity, and precipitation on the incidence of leishmaniosis and established different effects of individual climatic factors. In contrast to our study, they found that the incidence of leishmaniosis has a negative correlation with average yearly temperature and total yearly precipitation and a positive correlation with average yearly air humidity values [23]. In the study by Shirzadi et al., the authors examined the impact of meteorological components (temperature, evaporation, air humidity, and precipitation) on the incidence of cutaneous leishmaniosis, establishing a positive association of air temperature and evaporation with a rise in incidence, and negative association when relative air humidity and precipitation were concerned. The authors pointed out that climate changes before the active disease period (developed clinical picture of the disease) are a very important factor, with special significance if climate changes were observed for at least a month before the onset of symptoms of cutaneous leishmaniosis in people [29,40].

It is worth mentioning that the aforementioned temporal association may vary in geographical areas due to different seasonal patterns in different ecological zones [3]. Studies in French Guiana (South America) [41] and Rajasthan (India) [42] showed that the incidence of leishmaniosis increased with rising temperature, and decreased with reduced precipitation and lower air humidity. The spatial association of leishmaniosis and climate indicated that a changing climate may change or modify the spatial distribution or the incidence of this disease, although the pattern is specific to the studied geographical region. Climate variability may have different effects on disease spread and depends on the vectors in question and different Leishmania species in different geographical regions [43,44].

Many authors speak of the impact of climatic factors on the contraction of leishmaniosis through the impact of these factors on disease transmission vectors. The distribution of VL is significantly lower than the distribution of phlebotomus, since the disease transmission and distribution are based on vector concentration, their survival rates in an environment, transmission, and bite rate. Furthermore, it depends on the time of parasite incubation and the duration of the transmission period. All these parameters are based on climatic conditions. In places in which there has been a temperature rise due to climate changes, the disease prevalence is on a constant considerable rise [45].

Temperature and humidity play an important role in the survival, growth, and activity of sandflies. They are able to survive at lower temperatures in hibernation, initiated by the combination of low temperatures and shorter days, and which may last for 4 to 8 months, depending on the location. The temperature has also an effect on the activity development of the parasite. The worldwide distribution of sandflies is limited to regions with an average temperature of 20 °C, at least for a month. Sandflies are sensitive to sudden changes in temperature; suitable for them are usually the regions with a small difference between the minimal and maximal daily temperature. Their survival may be compromised if the climate becomes too warm and dry, although they may seek shelter in cooler and more humid spots during the day. Some species dwell in holes in the trees or logs. Peridomestic species may inhabit walls, and during the warmer part of the day they withdraw into the cracks in walls or rocks. Buildings made of porous rock provide favorable conditions for the development of sandflies since they collect humidity at nighttime which evaporates during the day. This suggests that climate has a direct impact on their development and thus, VL distribution as well, with an indirect impact on disease spread and reproduction of sandflies, i.e., distribution of reservoirs (hosts) and local vegetation (habitat). With climate changes, the areas of distribution of sandflies and Leishmania in appropriate developmental stage may spread to the north and higher altitudes. In the present endemic areas, higher seasonal temperatures would lead to prolonged periods of their activity and shorter periods of inactivity (hibernation). However, if the climatic conditions were too warm and/or dry, the vector survival rate would drop and the disease may disappear from some of the regions [39]. Higher temperatures during the day and humidity during the night have a strong impact on vector reproduction and distribution of leishmaniosis. In contrast, in countries with cold climate, the prevalence of the disease is limited since the development of sandflies is also limited (their activity is reduced or absent due to the inability to suck blood). A high level of average precipitation also has an important role in disease spread. Further, flooding may also spread the disease vectors and sandfly larvae to more distant locations where the disease has not been previously identified [45]. In the study by Dokhan et al., a direct association was established between open-air temperature and humidity with growing sandfly populations, while wind speed had an opposing influence [46].

A separate and combined analysis of the impact of climatic factors such as air temperature, humidity, and precipitation on phlebotomus in its habitats is especially important for a better understanding of their behavior and seasonal changes. The adaptability of flies to temperature and air humidity varies depending on the species in question and its bioclimatic distribution. Studies also show that changes in temperature, precipitation, and air humidity can have a strong impact on vector ecology through their distribution directions and impact on their survival and population size [47]. Soil and air humidity, as the consequence of rainfalls or soil saturation with water, are also important for sandflies; air humidity especially has an influence on their reproduction and life cycle [25]. Although little is known about the sites favorable for the reproduction of causative agents, for particular species, since a high percentage of humidity favors replication, the sites include wet soil in the Amazon region, caves, animal lairs, and certain human settlements [48,49]. Studies have shown that soil type and its saturation with water determine the distribution of sandflies [33,49]. The quantity of precipitation is a good parameter to determine humidity; it has been shown to play an important role in the efforts undertaken so far to model the distribution of leishmaniosis [34,35,49].

Seasonal and annual fluctuations of incidence are the principal characteristics of leishmaniosis epidemiology. These fluctuations are dependent on climatic factors, vector population dynamics, and host reservoirs, as well as on human behavior and migrations. Leishmaniosis is a disease sensitive to climate changes and precipitation; atmospheric temperature and air humidity changes have an impact in that regard. It is expected that the interplay of the effects of global warming and soil degradation will have an impact on the epidemiology of leishmaniosis through many pathways. Surveillance of the impact of climate changes and association of the disease incidence with seasonal fluctuations is therefore important for the control of the occurrence and recurrence of the disease, which may be adopted as an official measure in Montenegro. Although significant progress in medicine and science in general has been occurring during the past few decades, still there is a limited number of procedures that could be employed in order to prevent the spread of leishmaniosis. In the area of Montenegro, these might include insecticide application, treatment and immunoprophylaxis of animals, and increase in disease and general hygiene/nutrition awareness.

This study has several limitations since it does not provide information about the socioeconomical state of the diseased or collective immunity, nor about individual factors and risk-associated behavior such as increased exposure to sandflies during warm weather. In addition, some data regarding the disease report and appropriate diagnosis during the 70-year study period might not give a complete insight into the disease incidence over this long period.

## 5. Conclusions

Our study examined the impact of climatic factors (average temperature, air humidity, and precipitation) on the occurrence of leishmaniosis (or leishmaniasis) in Montenegro, excluding socioeconomic conditions, collective immunity, individual factors in the affected, etc. It is thus necessary to perform a study that would involve other factors in the environment, and socioeconomic characteristics of the population in the studied region, which may have an impact on the occurrence of this parasitic disease. Understanding of the role of ecologic and bioclimatic factors in the occurrence of leishmaniosis may provide guidelines for the creation of more effective programs for the prevention and control of this neglected parasitic disease.

## Figures and Tables

**Figure 1 ijerph-20-01656-f001:**
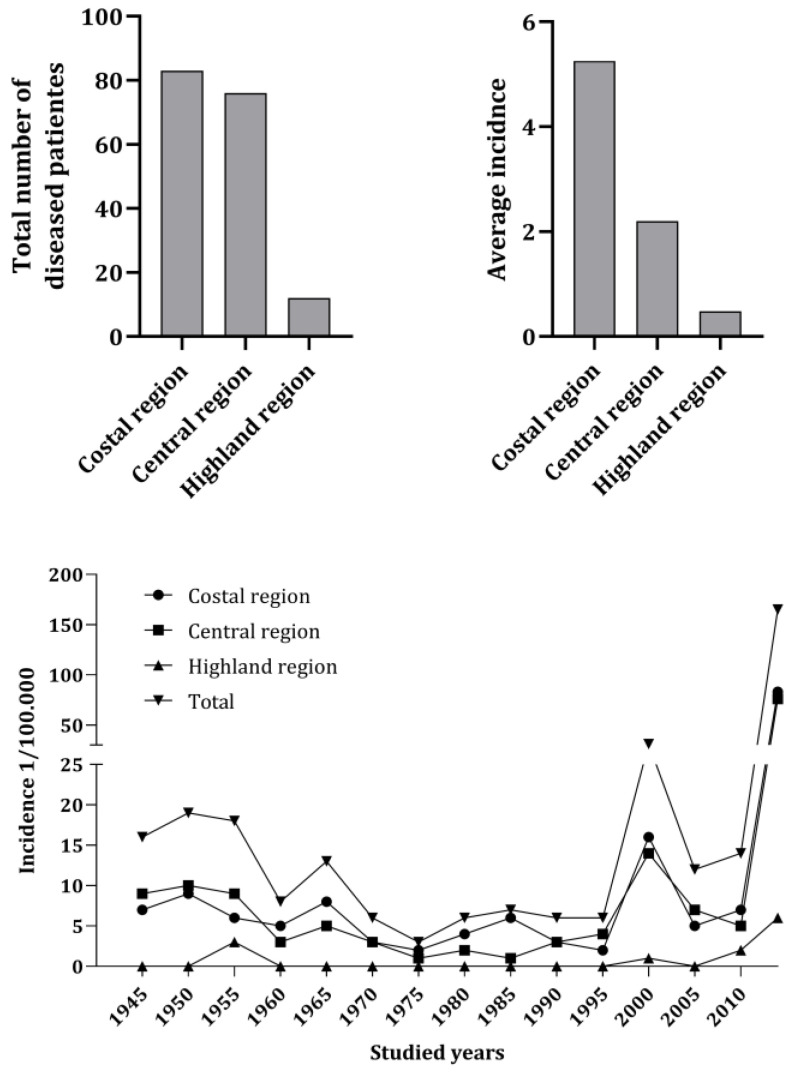
The number of the affected and average incidence rate of leishmaniosis by different regions in Montenegro in the period 1945–2014.

**Table 1 ijerph-20-01656-t001:** Association between the number of cases of leishmaniosis and air temperature (%) in coastal, central, and highland Montenegro.

	Coastal				Central				Highland			
	B	95% CI Cut-Off		*p*	B	95% CI Cut-Off		*p*	B	95% CI Cut-Off		*p*
		Upper	Lower			Upper	Lower			Upper	Lower	
**Temperature in January**	0.499	−0.652	1.649	0.389	1.016	0.416	1.616	**0.001**	0.499	−0.652	1.649	0.183
**Temperature in February**	0.161	−0.827	1.150	0.745	−0.017	−0.806	0.772	0.966	0.161	−0.827	1.150	0.469
**Temperature in March**	1.086	0.147	2.025	**0.024**	0.378	−0.588	1.344	0.437	1.086	0.147	2.025	0.469
**Temperature in April**	0.895	0.223	1.567	**0.010**	0.728	0.239	1.216	**0.004**	0.895	0.223	1.567	0.550
**Temperature in May**	0.514	−0.104	1.132	0.102	0.983	0.415	1.550	**0.001**	0.514	−0.104	1.132	0.805
**Temperature in June**	0.474	−0.020	0.968	0.060	0.631	0.268	0.994	**0.001**	0.474	−0.020	0.968	0.187
**Temperature in July**	0.530	0.057	1.003	**0.029**	0.702	0.318	1.087	**0.001**	0.530	0.057	1.003	0.117
**Temperature in August**	0.339	−0.122	0.801	0.146	0.538	0.216	0.861	**0.001**	0.339	−0.122	0.801	0.051
**Temperature in September**	0.338	−0.447	1.123	0.393	0.242	−0.242	0.726	0.322	0.338	−0.447	1.123	**0.026**
**Temperature in October**	0.098	−0.682	0.879	0.802	0.471	−0.098	1.039	0.103	0.098	−0.682	0.879	0.248
**Temperature in November**	0.260	−0.347	0.866	0.396	0.412	−0.029	0.852	0.066	0.260	−0.347	0.866	0.094
**Temperature in December**	0.428	−0.380	1.237	0.294	0.842	0.336	1.347	**0.001**	0.428	−0.380	1.237	**0.015**
**Yearly temperature**	1.068	0.117	2.019	**0.028**	1.326	0.652	2.000	**0.000**	1.068	0.117	2.019	0.060

**Table 2 ijerph-20-01656-t002:** Association between the number of cases of leishmaniosis and air humidity (%) in coastal, central, and highland Montenegro.

	Coastal				Central				Highland			
	B	95% CI Cut-Off		*p*	B	95% CI Cut-Off		*p*	B	95% CI Cut-Off		*p*
		Upper	Lower			Upper	Lower			Upper	Lower	
**Air humidity in January**	0.226	−0.063	0.515	0.123	0.044	−0.148	0.235	0.650	0.107	0.030	0.184	**0.007**
**Air humidity in February**	0.065	−0.080	0.210	0.371	0.141	−0.011	0.294	0.069	0.061	−0.009	0.131	0.086
**Air humidity in March**	0.225	0.021	0.429	**0.031**	0.245	−0.012	0.503	0.061	0.026	−0.038	0.090	0.422
**Air humidity in April**	0.003	−0.322	0.327	0.987	−0.226	−0.380	−0.071	**0.005**	0.002	−0.057	0.061	0.948
**Air humidity in May**	−0.107	−0.297	0.084	0.266	−0.317	−0.494	−0.140	**0.001**	0.131	0.045	0.218	**0.003**
**Air humidity in June**	−0.131	−0.315	0.052	0.158	−0.114	−0.219	−0.009	**0.034**	−0.024	−0.086	0.039	0.456
**Air humidity in July**	−0.056	−0.320	0.208	0.674	−0.168	−0.300	−0.036	**0.014**	−0.038	−0.094	0.019	0.186
**Air humidity in August**	−0.048	−0.358	0.262	0.757	−0.104	−0.208	0.000	**0.050**	−0.046	−0.084	−0.009	0.017
**Air humidity in September**	−0.714	−1.617	0.190	0.119	−0.123	−0.240	−0.006	**0.040**	−0.045	−0.100	0.011	0.113
**Air humidity in October**	0.271	0.070	0.473	**0.009**	0.041	−0.153	0.235	0.674	0.012	−0.122	0.147	0.853
**Air humidity in November**	0.022	−0.128	0.172	0.769	0.126	−0.086	0.338	0.238	0.184	0.058	0.311	**0.005**
**Air humidity in December**	−0.167	−0.395	0.062	0.150	0.935	0.277	1.594	**0.006**	0.057	−0.053	0.167	0.306
**Yearly air humidity**	0.099	−0.310	0.509	0.630	−0.251	−0.521	0.019	0.068	−0.012	−0.176	0.151	0.881

**Table 3 ijerph-20-01656-t003:** Association between the number of cases of leishmaniosis and precipitation (mm) in coastal, central, and highland Montenegro.

	Coastal				Central				Highland			
	B	95% CI Cut-Off		*p*	B	95% CI Cut-Off		*p*	B	95% CI Cut-Off		*p*
		Upper	Lower			Upper	Lower			Upper	Lower	
**Precipitation in January**	0.009	0.000	0.017	**0.042**	0.004	−0.002	0.009	0.189	−0.001	−0.005	0.003	0.603
**Precipitation in February**	0.017	0.004	0.030	**0.009**	0.012	0.004	0.019	**0.003**	0.009	0.002	0.017	**0.018**
**Precipitation in March**	0.015	−0.001	0.032	0.070	0.000	−0.008	0.008	0.978	−0.004	−0.012	0.003	0.232
**Precipitation in April**	−0.011	−0.032	0.010	0.306	−0.011	−0.019	−0.002	**0.013**	−0.003	−0.007	0.002	0.278
**Precipitation in May**	0.016	−0.010	0.043	0.226	−0.017	−0.044	0.009	0.202	0.014	0.005	0.023	**0.002**
**Precipitation in June**	0.035	0.011	0.059	**0.005**	0.034	0.009	0.059	**0.008**	−0.009	−0.016	−0.001	**0.030**
**Precipitation in July**	−0.016	−0.060	0.028	0.459	−0.022	−0.048	0.003	0.084	−0.005	−0.013	0.003	0.246
**Precipitation in August**	−0.035	−0.064	−0.005	**0.021**	−0.020	−0.038	−0.002	**0.032**	−0.007	−0.013	−0.001	**0.029**
**Precipitation in September**	0.005	−0.011	0.020	0.541	0.000	−0.010	0.010	0.972	−0.002	−0.006	0.002	0.353
**Precipitation in October**	0.007	−0.023	0.037	0.642	0.004	−0.004	0.012	0.287	0.000	−0.003	0.003	0.910
**Precipitation in November**	0.007	−0.009	0.023	0.378	0.000	−0.012	0.011	0.954	−0.005	−0.010	0.000	**0.031**
**Precipitation in December**	0.015	0.003	0.028	**0.018**	0.009	0.004	0.015	**0.001**	0.002	−0.002	0.005	0.355
**Yearly precipitation**	0.003	0.000	0.005	**0.027**	0.002	0.000	0.004	0.060	−0.001	−0.002	0.001	0.344

**Table 4 ijerph-20-01656-t004:** Association between the number of cases of leishmaniosis from 1945 to 2014 in Montenegro and yearly values of meteorological factors (air temperature, humidity, and precipitation).

Factor	B	95% CI Cut-Off		T	*p*
		Lower	Upper		
**Temperature (°C)**	**0.150**	0.013	0.287	2.166	0.031
**Humidity (%)**	0.031	−0.129	0.190	0.377	0.707
**Precipitation (mm)**	0.00005	−0.000002	0.001	1.911	0.057
**Constant**	**−** **4.039**				

**Table 5 ijerph-20-01656-t005:** Association between the number of cases of leishmaniosis from 1945 to 2014 in central parts of Montenegro and average yearly values of meteorological factors (temperature, humidity, and precipitation).

Factor	B	95% CI Cut-Off		T	*p*
		Lower	Upper		
**Temperature (°C)**	**1.178**	0.421	1.935	3.107	**0.003**
**Humidity (%)**	**−0.314**	−0.599	−0.029	−2.201	**0.031**
**Precipitation (mm)**	0.002	0.000	0.005	1.820	0.073
**Constant**	3.103				

**Table 6 ijerph-20-01656-t006:** Association between the number of cases of leishmaniosis from 1945 to 2014 in highland parts of Montenegro and average yearly values of meteorological factors (temperature, humidity, and precipitation).

Factor	B	95% CI Cut-Offs		T	*p*
		Lower	Upper		
**Temperature (°C)**	**0.174**	0.028	0.319	2.382	**0.020**
**Humidity (%)**	0.073	−0.101	0.248	0.840	0.404
**Precipitation (mm)**	−0.001	−0.002	0.000	−1.669	0.100
**Constant**	**−5.552**				

## Data Availability

The data presented in this study are available on request from the corresponding author.

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
