# Peer review of "Climate Variables Related to the Incidence of Human Leishmaniosis in Montenegro in Southeastern Europe during Seven Decades (1945–2014)"

_ijerph, 2023, doi:10.3390/ijerph20031656_

Round 1
Reviewer 1 Report
Although the topic is very interesting, it should be developed more clearly. I found the introductory part not very accurate. I would also better develop in the discussion what else has been published in the literature on the subject and add public health suggestions for improving the current situation.
Author Response
Although the topic is very interesting, it should be developed more clearly. I found the introductory part not very accurate. I would also better develop in the discussion what else has been published in the literature on the subject and add public health suggestions for improving the current situation.
Answer: The introduction section has been rewritten and new references have been added to the manuscript. The discussion has also been improved with new comments regarding the obtained results and some suggestion for improving current state are given. Also, tables in the supplementary part of manuscript further underline the conclusions of our work.
Reviewer 2 Report
I reviewed the manuscript tittle “Climate variables related to the incidence of human leishmaniosis in Montenegro in southeastern Europe during seven decades (1945-2014)” and manuscript number ijerph-2114225. The manuscript determined the association between temperature, air humidity, and weather conditions with the incidence of leishmaniosis in Montenegro during a seven-decade period (1945-2014). Considering climate and the changes associated with it, such as temperature change, precipitation, humidity, floods, salinity, and drought, have negative consequences on disease transmission, this study is necessary.
Here are some points that should be clarified:
1. In the background section, the authors should give an account of some literature that has carried out the influence of climate change on NTDs transmission e.g Schistosomiasis and climate change BMJ 2020; 371 doi: https://doi.org/10.1136/bmj.m4324 (Published 16 November 2020), Temperature and rainfall impact on schistosomiasis ISSN 0973-1768 Volume 14, Number 1 (2018), pp. 49–66, Estimating the Threshold Effects of Climate on Dengue: A Case Study of Taiwan Int J Environ Res Public Health. doi: 10.3390/ijerph17041392 etc.
2. The authors need to buttress the findings from their study with literature findings that reported the nexus between climate change and sandfly, because the authors have predicted that “Changes in the temperature, precipitation, and humidity can have major effects on vectors and host reservoirs, changing their distribution and influencing their survival and population size.
3. The authors need to state clearly each number of cases reported by each agencies or public health institutes and their year periods. Also, the meteorological that period obtained should be reported in the method section.
4. It will be very important to see the incidence data in each year in relation to the meteorological report of Montenegro over the period of years studied. Instead, the climatic weather data of Montenegro should be modeled and correlate with the incident data over the years studied. It wouldn’t be proper to base your leishmaniasis incidence over the period of 1945-2014 on just a year meteorological data. Perhaps that meteorological data is not a year, because I could see the yearly temperature, air humidity and precipitation reported in table 1, 2, and 3, respectively, it would be important to report your data based on each year. Check literatures for examples.
5. Follow up on the above comment, I am very sure that the climate data of the author studied area is not the same throughout for the period of years studies. For instance, the author wrote “Each precipitation increase by 1 mm was correlated with a significant rise in the number of people affected by leishmaniosis in coastal Montenegro in January, February, June, and December and at the yearly precipitation level by 0.003 (0.000 to 0.005; p<0.05), with a significant reduction of the number of cases in August – by 0.035 (-0.005 to -0.064; p<0.05). In central Montenegro, the number of cases of the disease significantly increased in February, June, and December, and significantly decreased in April and August”
6. It would be good to have both the meteorological and leishmaniasis incident data in the supplementary information.
Author Response
I reviewed the manuscript tittle “Climate variables related to the incidence of human leishmaniosis in Montenegro in southeastern Europe during seven decades (1945-2014)” and manuscript number ijerph-2114225. The manuscript determined the association between temperature, air humidity, and weather conditions with the incidence of leishmaniosis in Montenegro during a seven-decade period (1945-2014). Considering climate and the changes associated with it, such as temperature change, precipitation, humidity, floods, salinity, and drought, have negative consequences on disease transmission, this study is necessary.
Here are some points that should be clarified:
- In the background section, the authors should give an account of some literature that has carried out the influence of climate change on NTDs transmission e.g Schistosomiasis and climate change BMJ 2020; 371 doi: https://doi.org/10.1136/bmj.m4324 (Published 16 November 2020), Temperature and rainfall impact on schistosomiasis ISSN 0973-1768 Volume 14, Number 1 (2018), pp. 49–66, Estimating the Threshold Effects of Climate on Dengue: A Case Study of Taiwan Int J Environ Res Public Health. doi: 10.3390/ijerph17041392 etc.
Answer: The introduction section has been rewritten, new references (including the ones suggested by the reviewer) are added. Hopefully this section is in order now.
- The authors need to buttress the findings from their study with literature findings that reported the nexus between climate change and sandfly, because the authors have predicted that “Changes in the temperature, precipitation, and humidity can have major effects on vectors and host reservoirs, changing their distribution and influencing their survival and population size.
Answer: The discussion section has be extend and strengthened by contemporary references suggesting the association between sandfly and climate changes.
- The authors need to state clearly each number of cases reported by each agencies or public health institutes and their year periods. Also, the meteorological that period obtained should be reported in the method section.
Answer: The requested data about the number of cases has been added to the supplementary part of the manuscript. Also, official meteorological institute from which the data were obtain is added to the methods section.
- It will be very important to see the incidence data in each year in relation to the meteorological report of Montenegro over the period of years studied. Instead, the climatic weather data of Montenegro should be modeled and correlate with the incident data over the years studied. It wouldn’t be proper to base your leishmaniasis incidence over the period of 1945-2014 on just a year meteorological data. Perhaps that meteorological data is not a year, because I could see the yearly temperature, air humidity and precipitation reported in table 1, 2, and 3, respectively, it would be important to report your data based on each year. Check literatures for examples.
Answer: Indeed, the suggested way of data analysis and interpretation might give a different view point on the entire work. However, this study was conducted as a part of a previous doctoral study and this kind of statistical analysis would require a completely different approach and data presentation which at this point might not be plausible. Your suggestion would definitely be accepted since the presented issue of Leishmaniosis is a large project, that is an ongoing and this kind of data analysis might reveal some significant insight in the future.
- Follow up on the above comment, I am very sure that the climate data of the author studied area is not the same throughout for the period of years studies. For instance, the author wrote “Each precipitation increase by 1 mm was correlated with a significant rise in the number of people affected by leishmaniosis in coastal Montenegro in January, February, June, and December and at the yearly precipitation level by 0.003 (0.000 to 0.005; p<0.05), with a significant reduction of the number of cases in August – by 0.035 (-0.005 to -0.064; p<0.05). In central Montenegro, the number of cases of the disease significantly increased in February, June, and December, and significantly decreased in April and August”.
Answer: Indeed the change in climate is not the same through the year, however, single month/year changes would demand different statistical approach which was not conducted in the present study. Also, one should have in mind that this is a 70 year period of follow up that we presented here, and it would be quite complex to present this. The problem with discussion of this kind of data with a specific climate change of the month is the time of the disease report, which does not corelate directly with the time of infection (could be postponed for couple of months after the infection). This would bring a lot of confusion in the entire work.
- It would be good to have both the meteorological and leishmaniasis incident data in the supplementary information.
Answer: The table of the incidence was added to the supplementary, however the 70 year period in which the metheorological data of each month was collected would make a quite large table which would be completely useless since it would be impossible to organize. It mean that it would have at least 70 rows and 13 columns, not taking into account the area of Montenegro for which the data would be presented.
Reviewer 3 Report
This is an interesting and well-written article investigating statistical associations between climatic variables and the incidence of leishmanosis in humans based upon knowledge that the sandfly vector life cycle is dependent on climatic variables.
Although the authors do use appropriate terminology in terms of reporting the statistical interpretation of results as associations rather than causation and also mention that there are multiple potential methods of transmission of the parasite, more discussion about potential confounding factors is warranted. For example the interpretation that the differences are only climate driven may be a bit of a stretch, albeit interesting and warranting further investigation. Would it not also make sense that there could be compounding factors in terms of detection bias for an increased incidence in cases as well as an association with potential increased exposure to sandflies if people frequent the beach more during warmer weather?
Overall the paper is well-written, but would be improved with a bit more discussion about limitations of the study.
Line 56: should be "climatic changes" not "climate changes"
Line 298: the line "vectors and parasites" is confusing, re-word and be more specific what is meant here (sandflies and thus also the Leishmania stages?)
Author Response
This is an interesting and well-written article investigating statistical associations between climatic variables and the incidence of leishmanosis in humans based upon knowledge that the sandfly vector life cycle is dependent on climatic variables.
Although the authors do use appropriate terminology in terms of reporting the statistical interpretation of results as associations rather than causation and also mention that there are multiple potential methods of transmission of the parasite, more discussion about potential confounding factors is warranted. For example the interpretation that the differences are only climate driven may be a bit of a stretch, albeit interesting and warranting further investigation. Would it not also make sense that there could be compounding factors in terms of detection bias for an increased incidence in cases as well as an association with potential increased exposure to sandflies if people frequent the beach more during warmer weather?
Answer: Thank you for the comment, we added the suggested comments throughout the discussion section and in the part related to the study limitations.
Overall the paper is well-written, but would be improved with a bit more discussion about limitations of the study.
Line 56: should be "climatic changes" not "climate changes"
Answer: The error has been changed throughout the text.
Line 298: the line "vectors and parasites" is confusing, re-word and be more specific what is meant here (sandflies and thus also the Leishmania stages?)
Answer: This was corrected as requested.
Round 2
Reviewer 2 Report
The manuscript has been improved.